# D-CEWS: DEVS-Based Cyber-Electronic Warfare M&S Framework for Enhanced Communication Effectiveness Analysis in Battlefield

**DOI:** 10.3390/s22093147

**Published:** 2022-04-20

**Authors:** Sang Seo, Sangwoo Han, Dohoon Kim

**Affiliations:** 1Department of Computer Science, Kyonggi University, Suwon-si 16227, Korea; tjtkd8271@kyonggi.ac.kr; 2Ground Technology Research Institute, Agency for Defense Development (ADD), Daejeon 34186, Korea; swhan22@gmail.com

**Keywords:** cyber-electronic warfare, modeling and simulation, ad hoc, wireless threat, security

## Abstract

Currently, in the field of military modernization, tactical networks using advanced unmanned aerial vehicle systems, such as drones, place an emphasis on proactively preventing operational limiting factors produced by cyber-electronic warfare threats and responding to them. This characteristic has recently been highlighted as a key concern in the functioning of modern network-based combat systems in research on combat effect analysis. In this paper, a novel **d**iscrete-event-system-specification-based **c**yber-**e**lectronic **w**arfare M&**S** (D-CEWS) was first proposed as an integrated framework for analyzing communication effects and engagement effects on cyber-electronic warfare threats and related countermeasures that may occur within drones. Accordingly, for the first time, based on communication metrics in tactical ad hoc networks, an analysis was conducted on the engagement effect of blue forces by major wireless threats, such as multi-layered jamming, routing attacks, and network worms. In addition, the correlations and response logics between competitive agents were also analyzed in order to recognize the efficiency of mutual engagements between them based on the communication system incapacitation scenarios for diverse wireless threats. As a result, the damage effect by the cyber-electronic warfare threat, which could not be considered in the existing military M&S, could be calculated according to the PDR (packet delivery ratio) and related malicious pool rate change in the combat area, and the relevance with various threats by a quantifiable mission attribute given to swarming drones could also be additionally secured.

## 1. Introduction

Based on the modern warfare paradigm and the rapid change in the tactical ecosystem as a result of military modernization, ground forces are actively implementing modernization strategies [1] in order to maximize survivability and communication efficiency of various units by networking, intellectualizing, and mechanizing all sub-ground combat platforms. Agent-based modeling (ABM) [2] has recently been widely used to increase the reality and accuracy of various wargame models because it can extract actual results of phenomena from the autonomous judgments and actions of all platforms treated as agents in a complex combat system. In addition, the possibility of utilizing ABM as a major simulation element for analysis of combat effects according to wired and wireless communication within the concept of modernized network-centric warfare to quickly achieve both command and control (C2) according to layered command channels while sharing target and damage information between interconnected sub-combat platforms was demonstrated. The importance of information superiority based on rapid situation awareness was also proved with ABM.

### 1.1. Problem Statement and Functional Limitation in Tactical M&S

Currently, more attention is given to improving the efficiency of manned–unmanned operations through mission point-based surveillance patrol, deep strike, and visible and invisible firepower projection, or securing the integrity of communication channels through complete All-IP network-based overlay networking. In addition, the operational concept for preventing operational limiting factors of the ground forces from opposing forces’ specialized cyber-electronic warfare threats [3] in advance and responding quickly as an appropriate countermeasure was also not concretely considered [4] in current M&S. In other words, despite the fact that the concept of system operation is required in conjunction with the analysis of the effects in order to secure and maintain network security and restorability in tactical situations, related conceptual approaches and simulation analysis studies remain extremely insignificant both domestically and internationally. In addition, if the effects and logical modeling of the specialized opposing forces’ network and cyber-electronic warfare threat behavior is not sufficiently accompanied, countermeasures against each of the threats that may occur during actual operations cannot be prepared in advance. Accordingly, the total mission continuity and performance efficiency will be potentially reduced. Furthermore, if the attack-detection surfaces increase significantly as a result of the networking and intelligence of lower-level fighting platforms, it is projected that security maintenance measures should be prioritized.

### 1.2. Research Gap and Major Contributions

This paper proposes the use of the DEVS-based cyber-electronic warfare modeling and simulation framework (D-CEWS) for the first time as an integrated M&S framework for the analysis of communication and engagement effects against cyber-electronic warfare threats, which may occur within the dynamic swarm unmanned aerial vehicle system, to alleviate the aforementioned limitations and offer a more secure system. In addition, friendly blue forces’ countermeasures against all multi-layered jamming [5], man-in-the-middle attack [6], spoofing [7], DDoS [8], blackhole attack [9], wormhole attack [10], and network worm propagation and infection [11] cyber-electronic warfare threats were introduced by standardizing them as perfect Bayesian–Nash equilibrium (PBNE) and signaling game-based zero-sum competition strategies. In this case, the swarm unmanned aerial vehicle system entity modeled and simulated within the D-CEWS is representatively defined as an unmanned ground tactical vehicle (UGV) or quadcopter-type drone for reconnaissance and location reporting. The formulated and assigned responsibilities as a commander-level master node that communicates directly with the aerial/ground command and control center and a squad member-level slave node were also based on communication channels. The main mission scenario was conceptualized as a reconnaissance report, and the operational scenario of the unmanned system was abstracted as a path based on the master node depth and a single mission point and multiple waypoints.

For the first time, this paper addresses the limitations of officially reported studies [12,13,14,15,16,17,18,19,20,21] while presenting the DEVS-based M&S framework for the analysis of combat efficiency according to cyber-electronic warfare threats based on open source. In addition, the DEVS-based M&S framework can be practically first applied to conduct the analysis of effects of each potential cyber-electronic warfare threat on the military unmanned maneuver system. The system was mainly used within the ground combat platform, which was networked, mechanized, intellectualized, and smartened as a threat–response simulation analysis, based on all of the tactical communication-equipment specifications, operational tasks, and other tasks, and topologies. Furthermore, because the correlation between the mutual engagement efficiency of military agents following the threat’s incapacitation of friendly forces’ communication and the possibility of securing cyber security following the friendly forces’ application of the response logic was also calculated, an expansive framework was developed based on studies of the operation of the next-generation countermeasure against cyber-electronic warfare related to the response logic.

### 1.3. Paper Structure

The following is the structure of this article, which was aimed at proposing military M&S research based on cyber-electronic warfare. First, in Section 2, M&S studies related to our proposed work are investigated and evaluated. In Section 3, the proposed D-CEWS framework is defined, and a PBNE and signal game-based dynamic zero-sum game environment for optimizing response logics by threat is configured. In Section 4, the swarm and drone communication network for reconnaissance reporting is simulated as the main simulation entity in D-CEWS. The related cyber-electronic warfare threat types are made exclusive and classified based on the command communication flow and drone communication characteristics. The experimental results of cyber-electronic warfare threat and response logic are formulated in Section 5, which includes a battlefield scenario in which friendly and opposing forces engage each other, along with related communication and equipment parameters. Finally, in Section 6, conclusions are drawn based on the approaches and results.

## 2. Literature Review

Existing M&S studies related to the tactical environment were performed by various operational domain and related protocols due to the heterogeneous tactical network characteristics. At this time, the main goal of this study is to simulate the correlation between the communication environment change in the UAV according to the modeled cyber-electronic warfare radio threat and the engagement efficiency of the friendly forces. Accordingly, the scope of taxonomy is largely composed in this sub section of “agent-based battlefield M&S” research and “communication metric-based M&S” research.

### 2.1. Agent-Based Battlefield M&S

First, Hayward et al. [12], as part of a series of related agent-based battlefield M&S effect analysis studies, quantified combat effectiveness using three parameters: capabilities, environment, and missions, and rendered it intangible so that it could be used experimentally in a virtualized wargame experiment environment. Next, Cil et al. [13] proposed ACOMSIM for military use as a two-tiered hybrid agent architecture related to the analysis of small-scale engagement effects in a multi-dimensional future ground battlefield, thereby supporting the configuration of the directivity of quick command decisions of squad commanders based on structural inference. Seo et al. [14] proposed a multi-leveled structure of combat object models classified into platform and weapon concept levels based on the sub-divided DEVS structure concept. The authors tested this concept in abstracted small-scale engagement scenarios to show that it may be used in actual military wargames. They also demonstrated the directivity of expansion in simulations of combined operations-based large-scale conflicts in the future. Connors et al. [15] analyzed the effects of air combat based on key performance parameters specialized in the SACM (small advanced capabilities missile) weapon system, partially using agents and derived efficiency by parameters based on actual air combat mission scenarios. Thompson et al. [16] structured an agent-based modeling framework based on malicious code infection and propagation behavior to practicalize security analysis in MANET (mobile ad hoc network)-based mobile tactical networks. They compared the effectiveness of malicious code attenuation by a friendly forces’ defense approach to demonstrate the relevance of maintaining security in the military tactical environment.

### 2.2. Communication Metric-Based M&S

Shin et al. [17] presented an abstracted communication model in a network-centric warfare environment based on a propagation loss model in an engagement. The study proved the importance of promptness in decision-making for C2, signal attenuation, and gas loss based on the engagement efficiency increase and decrease ranges and the computational tradeoff. Furthermore, Li et al. [18] verified the modeling accuracy and ease of analysis of the decision point process for modeling the location of a cellular network base station and carried out a comparison based on major communication and decision metrics. Akhtar et al. [19] used the open-source-based micromobility modeling-based large-scale mobility tracking concept, real channel model, and calculated correlations and dependence based on all parameters, including vehicle flow and speed, received signal strengths, obstacles, and distance distribution, to simulate spatiotemporal changes in VANET (vehicular ad hoc network) topology. Based on communication and loss parameters, Regragui et al. [20] assessed the group mobility of commanders and squad-members-focused infantry formations grouped on MANET and determined more efficient maneuvering paths and communication stability for an agent. Lastly, Lee et al. [21] proposed an agent-based modeling process based on the correlation between the communication success rate and combat efficiency. The study verified the expansive possibility following the application of additional communication repeaters due to the inability of the commander to communicate in combat situations. 

### 2.3. Difference in the Proposed Framework

To alleviate and solve the abovementioned limitations described in Section 1.2, D-CEWS, is proposed, which is an agent-based cyber-electronic M&S framework based on PBNE and attack–defense state-based zero-sum game foreground and NS-3 communication simulator-based background. With each representative keyword, the analysis of the shortcomings of existing solutions and related differences between our works are also presented.

Full-scale modeling and simulation based on the cyber-electronic warfare: Most previously reported formal research only focused on communication effects and related engagement effects using a small set of communication indicators. In addition, concerning cyber-security subjects, studies have been performed based only on general domains that are unlikely to be realized in an actual military M&S communication environment, such as malicious payload-frame injection. However, the present study is based on the D-CEWS framework, simulating various wireless threats such as electronic warfare-based jamming, cyber-warfare-based routing attacks, and worm attacks, all of which are considered to be potentially high in the actual tactical environment, especially in the battalion-level dynamic wireless communication network. D-CEWS was also used to determine whether operational efficiency was enhanced or reduced as a result of communication issues per threat type by modeling this with the confidentiality, integrity, and availability (CIA) concepts.Configuration of countermeasures dedicated to each threat: The adaptive countermeasures of friendly blue defenders against various wireless threats such as jamming, man-in-the-middle attack and spoofing, DDoS, blackhole and wormhole-based routing attacks, and network worm propagation–infection threats were also standardized as zero-sum-based anti-jamming in the case of jamming. In addition, changes in the operational efficiency of the relevant blue forces were also simulated. This approach can be presented as an M&S idea that can solve the security limitations caused by heterogeneity in the military tactical network [22,23,24].Securing the practicality and reliability of M&S: Unlike other studies that have been officially reported, all of the parameters and derived results presented in this study were similarly standardized within the NS-3-based background with reference to the specification information operating in the actual battalion-level tactical unmanned system environment.Achieving game-theory-based optimization of attack and defense behavior: Previous studies have performed normalization and optimization of simulated results by repeatedly performing M&S or adjusting key parameters as statistically significant error values were applied. The optimization method based on the rewards for each executing assault and defense was standardized in this study by upgrading to a zero-sum-based game-theory context, further systematizing the approach. This approach was inspired by the direction of simulating the competitive behavior of actors by major threats in the wireless network [25,26].

## 3. Principles of Decision for the Proposed Work

The proposed D-CEWS framework is an optimized ABM simulation approach that effectively secures a communication and engagement effect analysis method for cyber-electronic warfare radio threats that were not reflected in previous M&S studies, and also alleviates both operational limitations and practical scalability issues. Therefore, in this section, the attack–defense actor model and related detailed components existing in the game modeling-based foreground and network simulation-based background included in D-CEWS are newly formalized by module, process, related tuples, and metrics.

### 3.1. Definition of D-CEWS (DEVS-Based Cyber-Electronic Warfare M&S) Framework

The main architecture of the proposed D-CEWS framework is shown in Figure 1. First, cyber-warfare-based threats, such as multi-layered jamming and anti-jamming, MITM and ARP spoofing, DDoS, blackhole and wormhole attacks, and network worm propagation–infection were applied in advance as potential cyber-electronic warfare wireless threats in the friendly swarm unmanned aerial vehicle system deployed in the battlefield. The correlations were given and advanced as attack agents. After that, the cyber-electronic warfare threat-based attack agents reflecting the unique characteristics by threat were used as interface components of the game modeling-based foreground and the network simulation-based background, respectively (① in Figure 1).

Definitions by actor model in the PBNE-based zero-sum game component were significant in the game modeling-based background. These definitions were used as preprocessed parameters for the actor’s dynamic signaling behavior. The defender’s “proactive and reactive tactics” in the “attack–defense state component” and the attacker’s “communication-equipment specification-based attack tactic composition”, for example, were variables to be used when displaying by metric in the verification module. All of these variables were used as significant behavior template concepts for each actor when performing agent-based M&S (② in Figure 1).

In addition, to potentially enable the occurrence of cyber-electronic warfare threats in network-centric warfare (NCW), a reconnaissance-combat operation scenario in the battlefield environment was composed. Elements for calculating the main cyber-electronic warfare threat simulation direction by swarm unmanned unit were parameterized and applied, based on the NS-3 simulator (③ in Figure 1).

### 3.2. Implementation of Zero-Sum-Based Dynamic Game Component

The game modeling-based foreground in D-CEWS comprises: (i) a zero-sum-based dynamic game component that was configured based on PBNE, and (ii) components to simulate both the threat and response logics for the next action by actor, based on the transition branch in NS-3 resulting in an attack–defense state including clustered elements and event information set groups. First, the actor’s attack and defense state components were configured as threat–defense models, formulated by the operational concept predefined in the cyber-electronic warfare-based battlefield scenario component within the NS-3-based background. The dynamic game component used the PBNE game strategy that optimizes the judgment that maximized payoffs for private asymmetry, based on incomplete knowledge by actors and by using detailed episodes. In addition, the quantitative sequential relationships were probabilistically applied for partial signaling between the opposing forces and the friendly unmanned defender, based on the reward concept. The leader and reactive follower-based Bayesian stochastic Stackelberg game (BSSG) strategies were also partially added by an actor in the PBNE. The detailed schematic layout described above can also be designed in detail based on 9-tuples [27], as illustrated in Figure 2, using a multi-layered jamming and anti-jamming-based electronic warfare competition concept.

$N=\left(N_{A},N_{D}\right)$ is the set of actors, $N_{A}$ is the opposing force, and $N_{D}$ is the unmanned defender. In this case, depending on the directivity of payoff, signaling, and feedback by an actor in a random episode, the opposing force was configured as an active cyber-electronic warfare threat leader and sender. The unmanned defender was configured as a passive, reactive follower and receiver, or, conversely, the unmanned defender was formulated as dynamic cyber-electronic warfare responding leader and sender. The opposing forces are formulated as passive naïve followers and receivers.$TS=\left(TS_{N_{A}},TS_{N_{D}}\right)$, $TS_{N_{D}}=\left(ts_{i}|i=1,2,\ldots ,n\right)$, and $TS_{N_{A}}=\left(\rho \right)$ are sets of actor types; $TS_{N_{D}}$ is defined as an element of the proactive or reactive event set-based private information of the unmanned defender $N_{D}$, and $TS_{N_{A}}$ is defined as an element of cyber-electronic warfare threat element set-based private information of the opposing force $N_{A}$. The types were either divided or combined with the abstraction based on the abilities to add and subtract rewards by actor, and the unmanned defender, which has a nondeterministic response logic set group, versus the opposing force decisively composing the element set with the intelligence validity of the unmanned defender, which is termed *ρ*.$GS=\left(GS_{N_{A}},GS_{N_{D}}\right)$, $GS_{D}=\left(gs_{N_{d_{i}}}|i=1,2,\ldots \right)$, and $GS_{A}=\left(gs_{N_{a_{j}}}|j=1,2,\ldots \right)$ are sets of game strategies related to mutual zero-sum competition between each opposing force $N_{A}$ and the unmanned defender $N_{D}$, which are composed based on the sender and receiver signaling relations. $GS_{D}$ is defined as a strategy group based on the proactive or reactive event sets possessed by the defender, and $GS_{A}$ is defined as a strategy group based on the sets of cyber-electronic warfare threat elements possessed by the opposing force as effective defender surface information.$SS=\left(SS_{N_{A}},SS_{N_{D}}\right)$, $SS_{N_{D}}=\left(ss_{N_{d_{i}}}|i=1,2,\ldots \right)$, and $SS_{N_{A}}=\left(SS_{N_{a_{j}}}|j=1,2,\ldots \right)$ are the sets of signals of the opposing force $N_{A}$ and the unmanned defender $N_{D}$, which are selected or deselected depending on the actor’s active or passive signaling mechanisms. The opposing force has $SS_{N_{A}}$ as the leading attack signal set to achieve invasion through the selected arbitrary cyber-electronic warfare threat, and the unmanned defender has $SS_{N_{D}}$ as a leading defense signal set for complete detection, avoidance, and prevention of the opposing force’s threat.$S=\left(s_{i}|i=0,1,\ldots k\right)$ is a set of finite states based on GS and SS in the game component and defines multi-level and transitivity in the cyber-electronic warfare threat–response environment along with actions.$A=\left(A_{N_{A}},A_{N_{D}}\right)$, $A_{N_{D}}=\left(a_{N_{d_{i}}}^{i}|i=1,2,\ldots x\right)$, and $A_{N_{A}}=\left(a_{N_{a_{i}}}^{j}|j=1,2,\ldots y\right)$ are finite sets of actions of the opposing force $N_{A}$ and unmanned defender $N_{D}$ for S. $A_{N_{D}}$ defines the defender’s detection, defending, or false negative actions for $s_{i}$ as a transitive relationship. $A_{N_{A}}$ defines the attacker’s actions for $s_{i},$ such as reconnaissance and search for the attack point of the opposing force on $s_{i}$.$\theta \left(S_{k},a_{x},a_{y},S_{k`}\right)$ is a probability distribution function to calculate the probability of reaching $S_{k`}$ when the opposing force $N_{A}$ performs the action referred to as $a_{x}$ and the unmanned defender $N_{D}$ performs the action termed $a_{y}$ in $S_{k}$ in the episode. $R\left(S_{k},a_{x},a_{y}\right)$ is a function that calculates rewards obtainable based on the judgment of the actor in the episode when the opposing force group $N_{A}$ and the unmanned defender $N_{D}$ perform the actions termed $a_{x}$ and $a_{y}$, respectively, in $S_{k}.$ Therefore, the actors compete in the direction to maximize reward.$U=\left(U_{A},U_{D}\right)$ is a signaling-based discount factor function, which cuts off the judgment ranges by an actor within [0, 1] to attenuate the effects of the signaling behaviors by an actor. Furthermore, with the limited views by the actor, it simulates the pre- and post-competitive strategy judgments by an actor in the leader–follower relationship along with the discrete flow of time.

As specified in the 9-tuples, the signaling performed by actors in D-CEWS was partially defined based on the signal game strategy to perform multistage-layer-based information transmission and BSSG. The causal relationship between transmission and reception was determined based on the directivity of optimization of payoff by actor and signaling initiative while being finalized as the PBNE-based optimized balance state.

In particular, the opposing force, as the sender and leader, transmits signals related to the initial reconnaissance or attack contact point search, based on malicious radio wave emission and unmanned system occupation to the unmanned defender model, in terms of cyber-electronic warfare threat-based attack behaviors by episode. Consequently, in response to the aforementioned, the passive unmanned defender, which represents the receiver and follower, dynamically performs actions such as detection, avoidance, and prevention based on the vulnerable contact point that did not consider prior defense or the preconfigured cyber-electronic warfare response logic concept. Thereafter, the opposing force assessed the validity and reliability of the defender’s intelligence and surface information possessed at the present time through the defender’s response to the previously transmitted signals. Based on these results, the opposing force reconstructed the invasion strategy and attack vector, amongst others, for the selected cyber-electronic warfare threat. In this case, the attacker’s judgment range was spatiotemporally cutoff according to a predefined discount coefficient, thereby causing both the reduction in the solution space and the derivation of an approximate value to calculate the optimal value of the total compensation related to the cyber-electronic warfare-based invasion target. In addition, based on private information to achieve the attack objective, the payoff was added to achieve asymmetry for the attacker’s advantage or, conversely, added for the advantage of the defender due to the wrong judgment of the attacker, and is finally calculated as utility, cost, and reward. Reward maximization through reasoning between the leading actors in the leader–follower relationship based on the signaling by episode was organized as Q-value optimization, as shown in Equation (1). In this case, U and TS are defined as actors with the signaling initiative in the current episode:(1)$$Q\left(S_{k},a_{x},a_{y}\right)=R\left(S_{k},a_{x},a_{y}\right)+U\sum_{S_{k`}}\theta \left(S_{k},a_{x},a_{y},S_{k`}\right)\cdot TS\cdot OPT\left(S_{k`}\right),$$

That is, $OPT\left(S_{k`}\right)$ from the viewpoint of the actively signaling opposing force is constituted as Equation (2) through SS, which is the signaling actions that can be performed in $S_{k`}$, and the maximum reward value optimized with an incomplete and private information-based judgment is calculated as follows:(2)$$OPT\left(S_{k`}\right)=\underset{SS}{\max}\underset{a_{x}}{\min}\underset{ay}{\sum }Q\left(S_{k},a_{x},a_{y}\right)\cdot \left(ss_{N_{d}{}_{i}}|i=1,2,\ldots \right),$$

Then, based on (1) and (2), PBNE was organized as in Equations (3)–(6), based on OD and OA. The entry into the corresponding equilibrium state was controlled according to the configuration of $U_{A}$ or $U_{D}$ based on whether the signaling leader was selected or not. In this case, $P_{D}$ in (3) is the prior probability-based judgment probability of the unmanned defender for $SS_{N_{A}}$ related $TS_{N_{D}}$, and $P_{D}^{\prime }$ in (4) is the posterior probability-based inference probability of the unmanned defender related to the $SS_{N_{D}}$ reconstructed and based on the updated internal detection and defense strategies after feedback-based signaling of the defender for $SS_{N_{A}}$.
(3)$$P_{D}=\left(p_{D}\cdot \left(TS_{N_{D_{i}}}\right)|i=1,2,\ldots n\right),$$
(4)$$P_{D}^{\prime }=P_{D}^{\prime }\left(\left(TS_{N_{D_{i}}}|i=1,2,\ldots n\right)|SS_{N_{A}}\right),$$
(5)$$OD(SS_{N_{A_{j}}})=arg\underset{SS_{N_{D_{k}}}\in SS_{N_{D}}}{\max}\underset{TS_{N_{D_{i}}}\in TS_{N_{D}}}{\sum }P_{D}^{\prime }\cdot F\left(TS_{N_{D_{i}}},SS_{N_{A_{j}}},SS_{N_{D_{k}}}\right),$$
(6)$$OA\left(TS_{N_{A_{i}}}\right)=arg\underset{SS_{N_{A_{j}}}\in SS_{N_{A}}}{\max}F\left(TS_{N_{A_{i}}},SS_{N_{A_{j}}},OD(SS_{N_{A_{j}}}\right)),$$

## 4. Construction of Cyber-Electronic Warfare Environment in D-CEWS

In this section, we define all of the preconditions, unique characteristics, and behavioral processes for identifying and M&S cyber-electronic warfare threat types that can potentially be caused within drone-based aerial unmanned maneuvering systems. It also formalizes the range of practical major attack–defense scenarios between swarm drones performing reconnaissance missions and enemy attack weapons.

### 4.1. Configuration of Unmanned Aerial System with Swarming Communication Drones

Based on a quadcopter-type drone, the features of the entity of the swarm-type unmanned aerial vehicle system in the battlefield environment to be primarily replicated in D-CEWS were defined.

Drones are aerial vehicles that fly using an autonomous navigation system that is operated remotely or via wireless input from afar without the need for a pilot. They belong to the category of unmanned aerial systems (UAS) together with unmanned aerial vehicles (UAV). The number of copters in the driving section, and the type of vertical takeoff and landing, are used to categorize them. In addition, drones are composed of sub-components, such as the “communication unit”, in which the GCS-based remote controller controls data transmission and reception; the “control unit” that controls the flight; the “driving unit” that performs flight actions; and the “payload unit”. Furthermore, by focusing on the “ground control station (GCS)” that provides actual mission data and communication equipment to deliver and remotely control commands, together with the “information providing device,” which has internal sensors and communication links so that environmental sensor information, such as optical, sound wave, direction detection, pressure, and visible light-infrared-thermal images, can be delivered and remotely controlled. These entities can be utilized according to the characteristics of the role and mission of the drones and can also possess structured communication processes in detail by protocol.

In addition, military-purpose drones are characterized by being operated mainly based on the intelligent surveillance and reconnaissance (ISR) system; the target acquisition, and reconnaissance (STAR) system; the joint surveillance target attack radar (JSTAR) system; the reconnaissance surveillance and target acquisition (RSTA) system; and the track and identify dismounted personnel (TIDP) system. In addition, for a weapon system type that project directly or indirectly firepower, military-purpose drones are designed to conduct unmanned combat aircraft-centered continuous-close air support (P-CAS), unmanned precision bombing and targeted strikes and C4ISR-based tactical information transmission/reception in parallel. Additionally, military-purpose drones are employed for nonweapon system reasons such as gathering opposing force intelligence, active evasion of radar detection systems, stopping the images of opposing force UAVs through cyber-attacks, and executing secret reconnaissance flights.

As a result, clustered multiple multi-hop wireless ad hoc based drones were formulated, based on master–slave within the D-CEWS scenario template, as shown in Figure 3, to perform cooperative reconnaissance reporting to friendly force ground commanders using computing vision sensor-based real-time communication, as per operation strategy.

### 4.2. Classification of Cyber-Electronic Warfare-Based Wireless Threat Types

Since drones rely on wireless communication networks, threats such as hijacking, service availability and data integrity damage, privilege escalation, and remote code execution may occur if security flaws are exposed or detailed components in a target are occupied. Ground-control devices and information-providing devices other than drones can also be targeted by specialized attackers [28,29,30].

As a result, threats such as multi-layered jamming, MITM, ARP spoofing, low-rate DDoS, blackhole assault, wormhole attack, network worm propagation, and infection were reorganized into cyber-electronic warfare threats in D-CEWS before being simulated, as illustrated in Figure 4. At this time, each of the presented wireless threats is described as follows within the simulated environment in the D-CEWS.

Multi-layered jamming: It is an electronic warfare attack to disrupt the radio communication behavior of the optimized friendly network based on various jammer types such as constant type and reactive type. It mainly performs information transmission and reception disturbances between targets in consideration of main radio wave characteristics such as a directional communication channel and frequency. At this time, these multi-layered jamming attacks can achieve service availability infringement and occupation of the target through the most simplified attack vector and decision logic. In addition, it can be formalized as an initial starting point to improve the possibility for success of electronic-protocol vulnerability-based exploits and more in-depth complex cyber-electronic warfare threat types, such as privilege escalation or side-channel attacks.MITM (man-in-the-middle) attack: It is a cyber-warfare attack in which a malicious attacker secretly penetrates a communication channel between a legitimate sender and receiver to eavesdrop, steal, steal, or modify packet information. This is also constituted as a representative logic for determining hidden intrusion and covert activity as a form of cyber-attack. At this time, these man-in-the-middle attacks can also be composed of a spoofing threat to be described, and a starting point vector that is basically applied to blackhole- and wormhole-based routing attacks.Spoofing: It is a cyber-warfare attack that bypasses the prescribed access control rules in the network in response to the request of a specific target as if it were a legitimate actor. It is mainly combined with MAC-based deception and masquerade attacks, GPS spoofing, and GNSS spoofing.DDoS (distributed denial-of-service) attack: It is a denial-of-service cyber-warfare attack that severely depletes the limited resources of the target and service specifications while neutralizing the target’s detection and blocking, and backtrack-based countermeasures to a certain extent. It is a logic that projects and pulses a number of illegal requests, mainly periodically or asynchronously and is also used as an artificial noise-inducing technique to safely perform an exploit-based deep attack vector while hiding it. In addition, this DDoS can also be formalized as a primary attack sequence that delays the target’s response so that other types of cyber-electronic warfare threats can succeed more easily.Blockhole attack: Unlike other cyber-electronic warfare threats, it is a threat type that is more specialized in an ad hoc network structure. It is a routing attack-based cyber warfare threat type that absorbs all packets by projecting false routing information from the attacker to the target node that has requested an optimized real-time routing path.Wormhole attack: It is an advanced, route-based cyber-warfare threat type based on the blackhole attack that performs a specialized routing attack on the target and evades the response system by using the shortest concealed tunnel amongst numerous cooperating bad actors to evade the response system.Network worm propagation–infection: It is a cyber-warfare attack that disrupts the overall operation of the target network by replicating itself at the malicious application level in a specific host and arbitrary service, then artificially propagating and infecting it through a communication protocol. At that time, unlike other cyber-electronic warfare threats, this attack was introduced as a type that could determine the specific attack logic at the only program level. Therefore, it can be standardized as an initial concept that can be used when constructing an in-depth attack logic or related detailed vector direction related to international security level standard concepts such as CVE (common vulnerabilities and exposures) and CVSS (common vulnerability scoring system).

The opposing forces always take the lead in threats of electronic warfare-based multi-layered jamming or cyber-warfare-based MITM, spoofing, DDoS, hole-based attack, and network worm propagation and infection, with the inventor and slave drones in the friendly forces’ unmanned military systems, which are made up of single or multiplexed drones, as the targets of the attacks. Additionally, since it simulates real-time cyber-electronic warfare threats within the wireless ad hoc based military unmanned maneuver system, all cyber-electronic warfare threats of the opposing forces were performed through spatiotemporal malicious radio waves. The concept of shooting range and radiation angle were also formalized so that they exist clearly as opposed to the existing wired legacy environment.

Multi-layered jamming, on the other hand, was chosen as an initial attack vector type that can be used as a starting point for exploits against other cyber-electronic warfare threats based on the following criteria, and sub-divided into random-type, constant-type, and reactive-type, among others, when configured [31].

As shown in Figure 5, the detailed jammer module related to multi-layered jamming is configured in detail based on the following requirements:The prior information and related intelligence required for the opposing forces to attack arbitrary master or slave drones in the swarm drone system, which is the target of an attack, should be minimized except for essential communication values (e.g., frequency, bandwidth, etc.), and the attack time should be short.Since the attack should be specialized as an availability disturbance attack based on a multi-hop wireless ad hoc based unmanned reconnaissance platform, the detailed attack vector should be configurable only based on the wireless network characteristics, excluding the brigade level or higher-level wired network characteristics.The execution process from the start of attack to the final success should be simple.The form of initial attack contact point interface should be provided as a ”starting point” so that successive attack chains can be formulated because the types of threats are not limited to the threat types buried in the scope of simple cyber warfare or electronic warfare, etc., but were fused into cyber-electronic warfare.It should also be easy to define friendly forces’ active–adaptive countermeasure-based interventions in opposing forces’ attack activities.

Accordingly, Figure 6 shows the jamming threat scenario associated with the examination of the operational efficiency effect associated with the reconnaissance report of the swarm communication drone maneuvering system employing multi-layered jamming in the D-CEWS.

Figure 7 depicts the anti-jamming reaction scenario based on the packet delivery ratio (PDR) and received signal strength (RSS) metrics of the friendly swarming maneuvering system’s master and slave drones to reduce the opposing red force’s jamming threat. Then, based on the following criteria, man-in-the-middle attacks and spoofing were chosen as the types of attacks that can compromise the confidentiality and integrity of reconnaissance data and packets in D-CEWS, and even partially damage confidentiality:They should be able to have specialized detailed attack vectors as confidentiality and integrity compromise attacks on multi-hop wireless ad hoc based unmanned reconnaissance platforms.They should be able to perform relevant exploit actions in a simple wired legacy environment and through radio wave radiation. In addition, they should be able to target any ground combat platform that is advanced based on All-IP.Beyond simply interfering with the sending and receiving of reconnaissance information of the friendly swarm drone system as with the multi-layered jamming threat described above, they must simulate the actions of the opposing forces’ agent who can read, take over, or modify the reconnaissance information.Legitimate existing sender-friendly drones and receiver-friendly commanders should not be able to immediately catch the relevant threatening actions, in contrast to the multi-layered jamming threats, which are immediately detectable with changes in the communication entropy based on metrics such as the packet delivery rate and received signal strength.Depending on the strength of the attack, the degree of damage to confidentiality, integrity, and availability should be dynamically changeable. Accordingly, the increase or decrease in operational efficiency should also be able to accompany.

Based on all these conceptual criteria, malicious men in the middle forcibly obtained legitimacy through camouflage and disturbance, they cause damage to confidentiality and integrity by taking over and modifying transmitted and received friendly forces’ reconnaissance information. In addition, both man-in-the-middle attack and spoofing threat scenarios were formulated, as shown in Figure 8 and Figure 9, so that communication–authentication service unavailability issues can also be partially caused.

Next, the distributed denial-of-service (DDoS), which is a type that can significantly damage the availability at the stacking stage of the All-IP-based wireless network communication protocol, rather than radio waves. The DDoS consumed only a small number of resource specifications and was also selected based on the following criteria:Existing jamming threats that cause availability disruption develop plenty of attack channels that the target can follow all the way back to the propagation terminal. It exposed many diverse residual artifacts (e.g., jammer location and size, jammer radiation range and three-dimensional movable range, etc.) through the analysis of the directivity of radiated radio energy. Accordingly, existing defenders can easily detect and neutralize it, either by reversely jammed or physically removed with invisible firepower projection. In order to minimize the reduction in attack and survival efficiency due to these countermeasures, it can be diversified and utilized as complex jammer types such as deceptive jamming and side-channel jamming. Eventually, the availability disturbance threat behaviors for the radio wave-based physical layer and the TCP and UDP-based network-transmission layer should be simulated to minimize the attenuation of the attack efficiency by the defender’s countermeasure actions, such as detection, evasion, prevention, and traceback. In addition, such threats should be able to target even the most advanced arbitrary ground combat platform based on All-IP.In the case of the multi-layered jamming threat described above, the variables per se, such as the radiation, period, and pattern of jamming energy, were divided by jammer type when they were operated. As a result, it should be able to create tradeoffs that benefit the attacker so that the limitation as a whole can be reinforced while the possibility of causing damage to the jamming type’s availability can be retained as much as possible.The attack should be optimizable as burst-based pulsing or constant-type attacks considering the target and communication environment at the wireless network protocol level.Similar to jammer threats, the degree of damage of availability should be dynamically and rapidly changeable according to the strength of the attacks, and it should be possible to accompany the increase or decrease in operational efficiency accordingly.

Based on these conceptual criteria, dead hosts were constructed that consumed a large amount of network resources while providing no significant artifacts to the unmanned defense. In addition, based on the aforementioned, the decrease in attack efficiency due to countermeasures such as immediate detection, prevention, and backtracking could be minimized, and the communication–authentication service unavailability issue in the swarm communication drone network could be greatly enhanced with the control of parameters advantageous to the attacker. Therefore, based on this approach, the DDoS threat scenario was composed, as shown in Figure 10.

Next, it should be possible to infringe confidentiality, integrity, and availability by conducting a more optimized denial of service attack based on ad hoc communication routing by performing disguise and disturbance within D-CEWS. In addition, the blackhole and wormhole attack, customized to consume only a small number of resource specifications, were selected based on the following criteria:The previously mentioned cyber-electronic warfare threats were not introduced as specialized threat types to perform optimum attacks in multi-hop-based dynamic wireless ad hoc environments, but were instead developed with the current legacy type of wired/wireless network–host topology in mind. Therefore, the threats that consider all the unique characteristics (collaborative routing and anomaly detection, node joining and leaving, etc.) cultivated by the wireless ad hoc network concept per se should be removed. The attack efficiency in comparison with existing threats should also be improved.It should be able to possess specialized, detailed attack vectors as attacks that cause damage to the confidentiality, integrity, and availability of multi-hop wireless ad hoc based unmanned reconnaissance platforms. It should also be able to exploit actions centering on wireless ad hoc. In addition, it should be able to target any ground combat platform that was advanced based on the All-IP environment.For changes in network transmission and reception based on perturbation, viewing and hijacking, unauthorized modification, etc., legitimate existing sender-friendly drones and receiver-friendly commanders should not be able to easily catch the threat.The concept of multiple malicious collaborators should be considered, and a complex attack chain against it should also be configurable.

The efficiency of dedicated attacks in a wireless ad hoc based swarm communication drone network was optimized using camouflage and disturbance actions based on the projection of false optimal routing information and node role information based on all of these fundamental criteria. The threat scenarios for blackhole attacks and wormhole attacks were also formulated using the mentioned criteria, as shown in Figure 11 and Figure 12, respectively, so that even the challenges of availability, integrity, and confidentiality damage could be greatly improved with attacker-controlled parameters.

Finally, the network worm propagation and infection attacks that caused the most extreme availability disturbance threat within the friendly swarm communication drone network as a threat compatible with the application layer to perform more advanced denial of service attacks within D-CEWS were also selected based on the following criteria:The multi-layered jamming and DDoS, which were predefined to cause a threat to the availability within the friendly swarm communication drone system for reconnaissance reporting that was turning around, have a limitation that the opposing force should always be projecting directional radio waves or network access to the friendly drone system without fail. That is, if the opposing forces are unable to gradually improve attack efficiency due to the friendly drone system’s immediate response actions, the longer the attack duration, the greater the chance of exposing meaningful artifacts to friendly forces and being targeted as an object of traceback and invisible firepower projection. Accordingly, an advanced attack type in the form of APT (advanced persistent threat) should be created and presentable from the perspective of M&S so that threats to the friendly forces can be automatically carried out at the application level with only one successful attacker-led invasion and exploitation.It should be possible to construct a kill chain that can maximize the induction of the target’s cognitive bias to be compatible with the more sophisticated social engineering attack concept for military operating environments such as disinformation and deception.Beyond simply occupying friendly networks, it should also be able to take over control.It should be possible to achieve the threat state and the occupation of the target node more quickly.

Based on all these conceptual criteria, a network worm threat scenario was finally configured, as shown in Figure 13, so that traffic and resources were highly consumed through self-replication in dynamic ad hoc and repetitive creation of socket communication while the highest availability damage issue compared to existing threat types can be realized with the control of SIR (susceptible–infectious–recovered) model [32,33]—based parameters advantageous to the attacker.

## 5. Simulation and Results

Based on all of the configured friendly and enemy entities, cyber-electronic warfare wireless threat process and related engagement scenarios, in this section, all major experimental arguments in D-CEWS are configured with NS-3 by tactical operation schemes and related threat types. According to standardized parameters and threat types, PDR-based communication effect and remaining combat ratio-based combat effect analysis are performed as a whole, and related graphs optimized with independent true random generation-based Monte Carlo processes are finally calculated.

### 5.1. Modeling of Battlefield Scenarios and Related Parameters

The core of this simulation was that the cyber-electronic warfare threat of the specialized opposing force for the master–slave-type swarm drone maneuver system that performed cooperative reconnaissance reporting in the battlefield environment on a bottom-up basis was assumed. The effect of decreased reconnaissance reporting efficiency, owing to the threat outbreak on the efficiency of combat missions that could occur between friendly and hostile troops, was studied in particular. The D-CEWS framework was schematized from M&S perspective in this case, comprising the zero-sum-based game background and the NS-3-based network simulation background, as illustrated in Figure 14.

The opposing force attacking using the cyber-electronic warfare threat type has a process to disturb the radio waves radiated by a swarm drone turning around at low altitude sky in order to perform a reconnaissance report on the enemy’s line to a friendly commander with threat vectors in the radio waves. The opposing forces mainly disturb the communication channel availability for reconnaissance data or incapacitate the transmission and reception scheme through multi-layered jamming, DDoS, hole-based attack, network worm propagation–infection attack, amongst others. The opposing forces can also damage confidentiality and integrity with the theft, hijacking, and unauthorized modification of packets through man-in-the-middle attacks, spoofing, amongst others.

In response to the foregoing, the friendly forces’ drone maneuvering system can also be defined as an optimization process to maximize reconnaissance operations’ efficiency. Based on major communication metrics such as packet delivery ratio (PDR) and received signal strength (RSS), the friendly forces can detect cases where the internal robustness and security values of wireless communication were reduced below the certain threshold values due to the intervention of certain malicious actors (i.e., opposing forces that project specialized cyber-electronic warfare threats) and performed dynamic countermeasures to avoid or prevent such cases. 

Table 1 shows the communication and functional simulation parameters of the friendly forces’ communication drone system that turned around to execute such reconnaissance reporting within the D-CEWS in this case.

**Table 1 sensors-22-03147-t001:** D-CEWS (DEVS-based cyber-electronic warfare)-based communication and functional simulation parameters for reconnaissance report communication simulation of swarming drone system.

Parameter (1/2)	Value	Parameter (2/2)	Value
Simulation time (s)	100~1000	Delay model	Constant speed propagation
Number of runs	10	Loss model	Friis, TwoRayGround
Size of battlefield (m)	1000 × 1000~3000 × 3000	Mobility model	Constant position
Number of slaves	10	TCP/IP stack	IEEE 802.11b
Number of master and GCS	1	Transmission power (dBm)	5~47
Channel model	DSSS, OFDM (WNW [34])	Packet interval (s)	0.01~1
Channel capacity (Mbps)	1~11	Guard interval (ns)	1600
Bandwidth (Mbps)	0.128~10	Tx gain, Rx gain (dB)	−1, −10
Frequency (MHz)	22	Routing protocol	AODV
Packet size (byte)	32~1024	Authentication algorithm	ECDSA-based
Velocity of drones (m/s)	2~10	Number of mission points	1
PER and BER reference	NIST model	Engagement distance (m)	100~1000
Main mission	Reconnaissance report	Behavior for main mission	Communication relay

Among the communication and functional simulation parameters in D-CEWS related to the cyber-electronic warfare-based execution of each threat, parameters that were representatively related to multi-layered jamming and anti-jamming are shown in Table 2.

In addition, all MITM and ARP spoofing, DDoS, blackhole attack and wormhole attack, and network worm propagation–infection attacks were performed as attack simulations of the opposing forces based on the parameters in Table 1. In addition, the defined cyber warfare wireless threats are determined based on Table 3 and constitute a competitive attack process.

Finally, the definitions by unit for the battlefield environment to perform wargame M&S, based on Figure 14, were finally configured, as shown in Table 4. In this case, the probability of detection (PoD) indicated the probability of detecting the opponent, and the probability of hit (PoH) was a performance value following the execution of virtualized shooting actions.

Additionally, as each threat was performed in the M&S testbed based on Table 4, classified elements of damage effect, as shown in Table 5, were also configured.

### 5.2. Experimental Results

The swarm communication drone maneuvering system was targeted because it performs reconnaissance reporting on the position of hostile forces’ combat units in order to increase allied combat squads’ non-line-of-sight (NLOS) strike accuracy. It was an M&S scenario in which the cyber-electronic warfare threat of the opposing force occurred, and the Monte Carlo based iterative simulation results were related to the D-CEWS-based communication effects, and engagement effects were calculated and analyzed. In this case, according to Table 4, the simulation ends when the combat power of the friendly forces, which has twice as many combat units as the opposing force, is reduced by 50% or more compared to the initial combat power, or when the combat power of the opposing force is reduced by 70% or more. After that, the remaining combat power at the time of completion was formulated with experimental results. The blue remaining combat ratio (BRCR) and red remaining combat ratio (RRCR), which are the measure of effectiveness (MOE) [21], related to the combat powers by squad, are defined by Equation (7):(7)$$\mathrm{BRCR}=\frac{B_{e}}{B_{s}}\cdot 100,\;\;\;\mathrm{RRCR}=\frac{R_{e}}{R_{s}}\cdot 100,$$
where $B_{e}$ is the combat power of the friendly forces at the end time, $B_{s}$ is the initial combat power of the friendly forces at the start time, $R_{e}$ is the combat power of the opposing force at the end time, and $R_{s}$ is the initial combat power of the opposing force at the start time. In addition, the packet error rate (PER) and the signal-to-noise ratio (SNR)-based loss models, dynamic mobility models, jitter, and latency, etc., in D-CEWS were customized and the NS-3-based class was used, as presented in Table 1.

First, Figure 15 is a set of normalized results for the reduction in PDR of the reconnaissance report drone and the BRCR of the related friendly forces following the electronic warfare-based reactive jammer-based multi-layered jamming attack. Based on this result, it can be shown that the engagement efficiency of friendly combat squads was greatly reduced following the incapacitation of the drone reconnaissance reporting communication of the reactive jammers of the opposing forces gradually radiated after warm-up for seven seconds. In addition, it was shown that the BRCR, which gradually decreased as the PDR of drones decreased, was also finally fixed at 53.5% when the PDR was about 30%.

This trend was based on all of the shutdown and availability damage of the friendly reconnaissance report communication channel due to the strong jamming power, or the loss and integrity damage of the transmitted and received reconnaissance information due to the somewhat weak jamming power. The DEVS-based M&S proved that this had a negative impact on friendly combat squads performing invisible strikes by collecting the position of each combat unit of the opposing force from the commander in real-time.

Figure 16 is reactive response logic for the opposing force’s reactive jammer-based multi-layered jamming attack. This was the PDR and BRCR-based result set normalized for the dynamic application of a channel hopping-based anti-jamming scheme to all sub-ordinate slave drones by the only master drone in the friendly forces’ swarm communication system that performed reconnaissance reporting when jamming actions were detected based on the PDR and RSS-based thresholds. For that, it was a normalized PDR- and BRCR-based result set. The engagement efficiency of friendly combat squads can also be improved by introducing the anti-jamming response logic, which guarantees the availability and integrity of drone communications for reconnaissance reporting damaged by the opposing forces’ jamming attack at least to a certain level, according to the relevant results derived from an electronic warfare-based simulation of this zero-sum-based competitive relationship as such. If the engagement was started with an equal share relationship, despite the change in the adaptive reactive jammer of the opposing force, the opposing force jammer with limited fighting power and limited radiation-distance–radiation-angle would no longer maintain the asymmetric superiority according to the spatiotemporal changes.

At this time, the “channel hopping”-based anti-jamming technique defined has proven to have a favorable effect on friendly combat squad engagement by communicating and receiving reconnaissance report data through a mutation-based communication channel that the enemy’s jammer cannot recognize. Accordingly, in order to meet the needs of the military to realize proactive defense by applying the concept of active avoidance based on cyber-agility [35,36,37], the optimal adaptive communication mutation method within DEVS-based M&S may also be adopted.

Following the cyber-warfare-based MITM attack by the opposing forces, Figure 17 illustrates sets of results normalized according to the number of men-in-the-middle for reducing the PDR of the reconnaissance report drone and the BRCR of the related friendly forces. The relevant results damaged the confidentiality by reading and taking over the reconnaissance information transmitted in the channel with a disguise or disturbance. To compromise the integrity of the data, the information was altered without authorization or turned into disinformation. Furthermore, the availability was damaged by preventing all friendly forces from transmitting or receiving reconnaissance information per se with gradual theft and intervention. Therefore, as the number of men-in-the-middle increased, the robustness of the friendly forces’ reconnaissance reporting was also lost in multiple stages based on the PDR. It can be shown that the engagement efficiency of the friendly forces also decreased. This is similar to the attack simulation of the opposing forces based on another camouflage and disturbance such as ARP spoofing or wormhole attack.

Furthermore, Figure 18a,b shows the result sets normalized according to the number of infected drones for the reduction in the PDR of the reconnaissance report drone and the BRCR of the related friendly forces following the cyber-warfare-based network worm propagation and infection attacks by the opposing forces. Through the relevant results, extreme traffic was generated through self-replication in MANET and repeated socket communication creation, and network resources were also highly consumed. As a result, the availability of the friendly drone’s clustered communication networks per se was completely damaged, and it could also be shown that the engagement efficiency was reduced due to the inability of the friendly forces to report the reconnaissance.

Figure 19a,b shows the result sets of PDR and BRCR for blackhole and wormhole based routing attacks. Through the corresponding result, the change in engagement efficiency according to the packet absorbing concept possessed by blackhole and wormhole attacks was confirmed with relationship PDR and BRCR. It was also shown that wormhole attack based on the cooperative malicious behavior between multiple attackers is a more effective threat in swarming drone network than blackhole attack.

Finally, Figure 20a,b shows the result sets of PDR and BRCE according to the DDoS pulsing rate and the number of zombies. From the result, it was confirmed that the minimized PDR and BRCR were derived from the DDoS pulsing rate of 8 Mbps or higher, and it was also shown that the DDoS attack efficiency was maximized from 50 or more zombies.

## 6. Discussion

This study proposed the D-CEWS framework for the first time to analyze the communication effect and related engagement effect based on the cyber-electronic warfare threat generated by the master and slave drone-based swarm unmanned maneuvering system that performed cooperative reconnaissance reporting to the commander based on the DEVS. In addition to electronic warfare-based multi-layered jamming, which could be used independently or in combination with other cyber warfare threats, potential attacks that were highly likely to be induced during the execution of operations in the tactical operation environment, such as cyberwarfare-based MITM, spoofing, DDoS, blackhole attack and wormhole attack, network worm propagation, and infection attack, etc., were analyzed. In particular, the opposing forces’ attack logic and the friendly forces’ response logic for this type of attack contact points and threats were simulated in detail as PBNE and signal game-based zero-sum game competition strategies and formulated as a cutoff scenario. Plans to secure the security of wireless communication measures that must be actually guaranteed by ground unit corps that want to improve engagement efficiency with collaboration between reconnaissance drone-based intelligent ground platforms introduced in accordance with the military’s modernization could be considered through the cyber-electronic warfare-based wargame M&S. In addition, it was possible to derive the quantitative correlation between the engagement efficiency according to the communication metric value changed following the execution of the cyber threat or countermeasure and the direction of optimization of the response logic of the related friendly forces. However, the following threats to the validity of this study have been identified.

Scalability and diversity issues: All the derived results of the simulation are limited to showing the communication effect and engagement effect on the cyber-electronic warfare threats that occurred within the unilateral C2 channel from the perspective of the drone performing the reconnaissance report and the entire squads centered on the commander. That is, the M&S according to threats to platforms other than drones should be considered, and detailed simulation effect analysis by combat unit performing actual engagement should also be performed. Furthermore, in addition to the PDR and RSS-based experimental results, additional communication metrics such as PER, BER, throughput, bandwidth, reliability, latency, load, routing, and resource should also be considered to conduct effect analysis. Variables specialized in the cyber-environment such as attack success probability, defense success probability, transmission success probability for reconnaissance, and measure of effectiveness for combat, etc., should also be considered.Reliability and practicality issues: Although this study is a DEVS-based M&S study that considers the complexity related to cyber-electronic warfare threats in the rapidly changing tactical network operating environment following military modernization, the experimental results cannot represent the actual battlefield environment due to the nature of the research field. Accordingly, it is necessary to secure the reliability of the model through additional verification routines such as augmentation and normalization.Issue of quantitative comparison to previous studies: According to the authors’ maximum investigation and judgment, M&S studies on military communication effects and combat effects related to cyber-electronic warfare threats have not been officially reported. That is, it is difficult to quantitatively compare the proposed studies with similar research fields by attribute. Since this situation should be due to the fact that proposed this study based on D-CEWS first identified cyber-electronic warfare threats not considered in previous studies and applied the threats to a virtualized warfare environment to simulate the effect analysis in order to fully secure the distinction of this study, it will be necessary to additionally derive simulation results related to the definition of meaningful tactical operation scenarios.

## 7. Conclusions

In this study, a networked and smart drone-based swarm unmanned maneuvering system security simulation study was proposed to analyze the M&S effect related to the cyber-electronic warfare threat and response logic within the modernized military wired-wireless networks. In addition, it was formulated in detail as a D-CEWS framework. Through the foregoing, with a basic analysis of threat and response simulation based on all of the actual military specifications, operational tasks, topology, etc., the practical utilizable as cyber-electronic warfare threat–response M&S for multi-layered jamming, man-in-the-middle attack, spoofing, blackhole attack and wormhole attack, and network worm propagation and infection threats has been proven with the results of experiments. In addition, the correlations between the mutual engagement efficiency by military agents following the incapacitation of friendly forces’ communication by threat and the possibility of securing cyber-security following the application of the friendly forces’ response logic were also derived by major metrics. In the future, this study will be expanded to studies on the operation of next-generation cyber-electronic warfare countermeasures related to the response logics by threat in various unmanned operation platforms and M&S S/W optimized for wargame engagement situations in the ground-force environment, which is actively modernized along with military modernization, and dedicated to cyber-electronic warfare threats will be concretized.

## Figures and Tables

**Figure 1 sensors-22-03147-f001:**
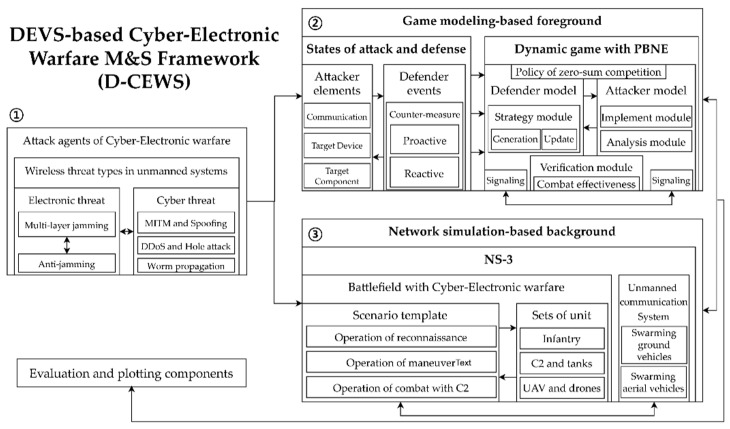
Main overview of D-CEWS (DEVS-based cyber-electronic warfare) framework.

**Figure 2 sensors-22-03147-f002:**
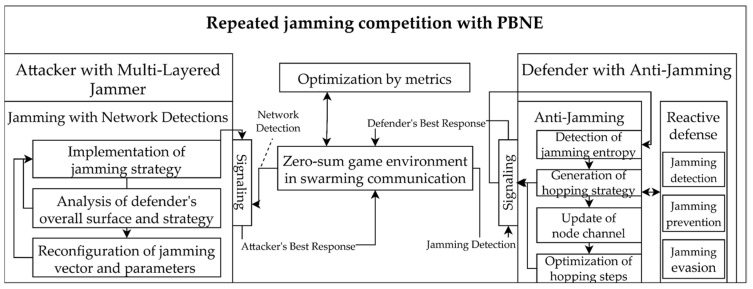
Detailed sub-overview of the zero-sum-based dynamic cyber-electronic warfare competition with PBNE (perfect Bayesian–Nash equilibrium) in jamming threat from D-CEWS.

**Figure 3 sensors-22-03147-f003:**
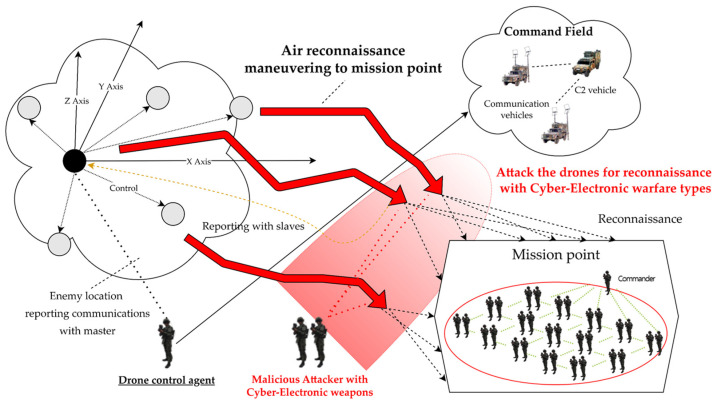
Sub-overview of swarming communication blue drones for reconnaissance in D-CEWS (DEVS-based cyber-electronic warfare).

**Figure 4 sensors-22-03147-f004:**
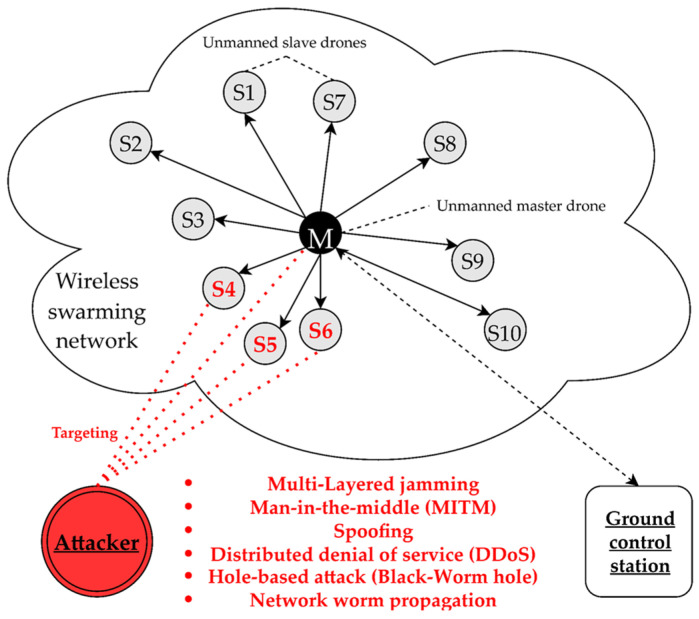
Sub-overview of cyber-electronic warfare for swarming drones in D-CEWS (DEVS-based cyber-electronic warfare).

**Figure 5 sensors-22-03147-f005:**
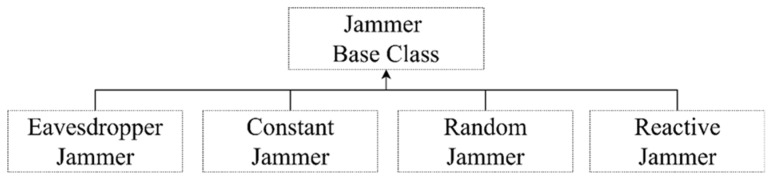
Detailed jammer types within the jammer component.

**Figure 6 sensors-22-03147-f006:**
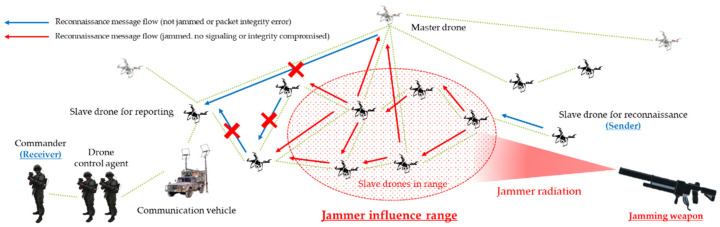
A multi-layered jamming scenario for a reconnaissance report in a swarming communication drone maneuvering system.

**Figure 7 sensors-22-03147-f007:**
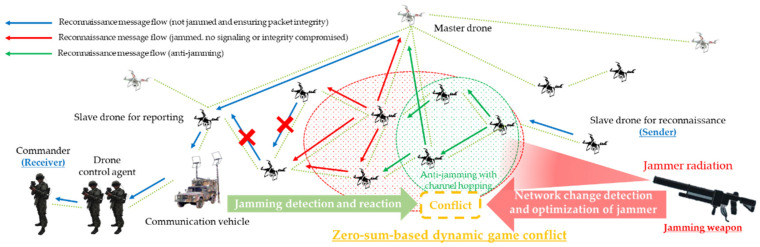
Anti-jamming scenarios for multi-layered jamming that interfere with reconnaissance reports within a swarming communication drones maneuvering system.

**Figure 8 sensors-22-03147-f008:**
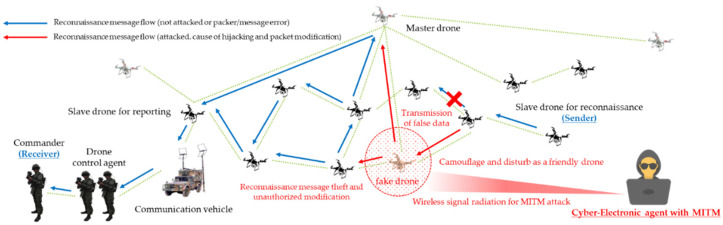
MITM (man-in-the-middle) attack scenario for reconnaissance report in a swarming communication drones maneuvering system.

**Figure 9 sensors-22-03147-f009:**
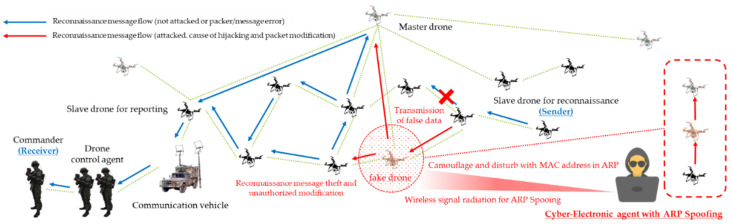
ARP (address resolution protocol) spoofing scenario for reconnaissance report in a swarming communication drones maneuvering system.

**Figure 10 sensors-22-03147-f010:**
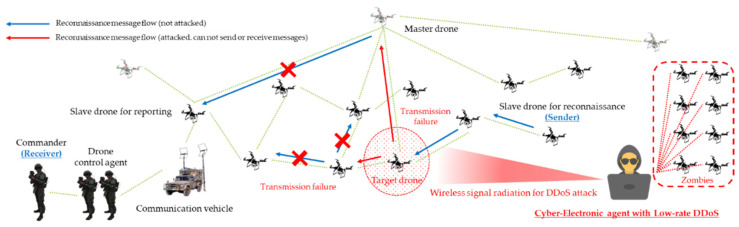
DDoS (distributed denial-of-service) scenario for reconnaissance report in a swarming communication drones maneuvering system.

**Figure 11 sensors-22-03147-f011:**
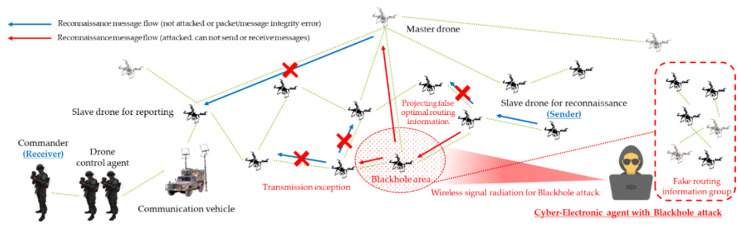
Blackhole attack scenario for reconnaissance report in swarming communication drones maneuvering system.

**Figure 12 sensors-22-03147-f012:**
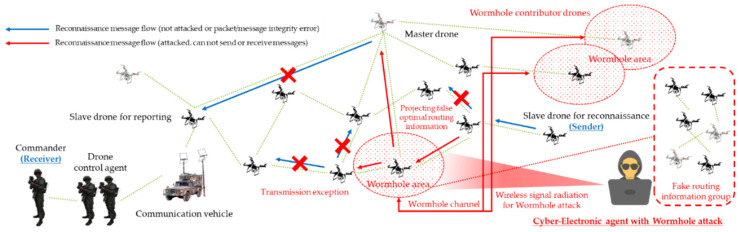
Wormhole attack scenario for reconnaissance report in a swarming communication drones maneuvering system.

**Figure 13 sensors-22-03147-f013:**
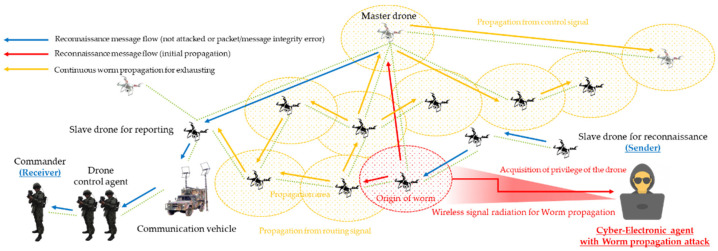
Network worm propagation and infection scenario for reconnaissance report in a swarming communication drones maneuvering system.

**Figure 14 sensors-22-03147-f014:**
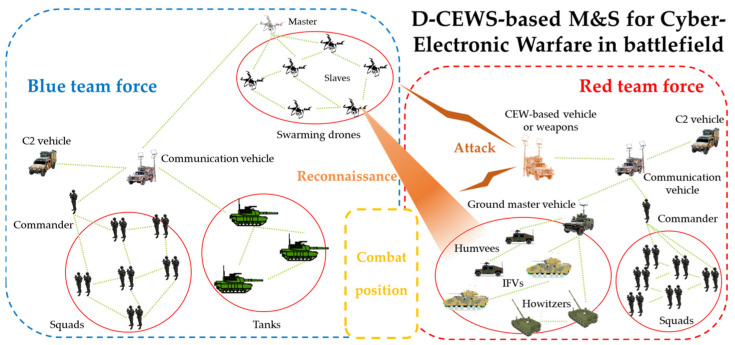
Main overview of D-CEWS (DEVS-based cyber-electronic warfare)-based modeling and simulation scenario for cyber-electronic warfare with reconnaissance report in a swarming communication drones maneuvering system.

**Figure 15 sensors-22-03147-f015:**
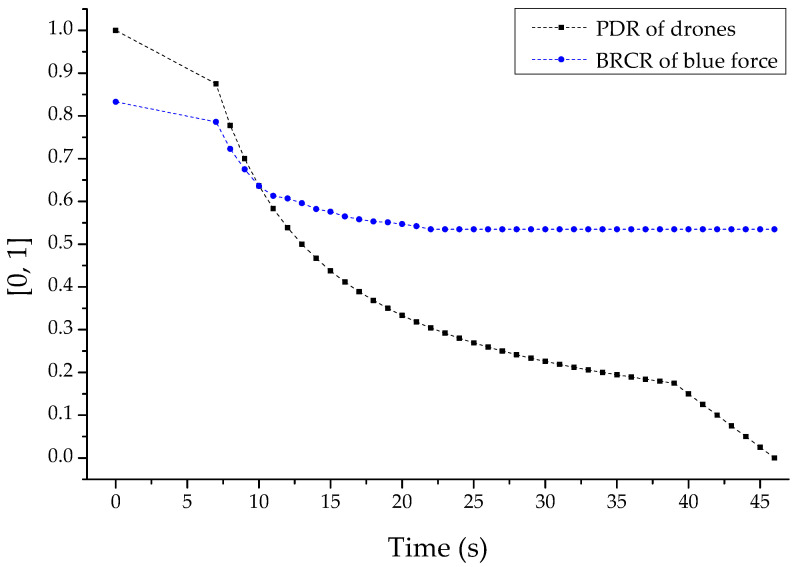
PDR (packet delivery ratio) and BRCR (blue remaining combat ratio)-based graph by reactive jammer-based multi-layered jammer attack.

**Figure 16 sensors-22-03147-f016:**
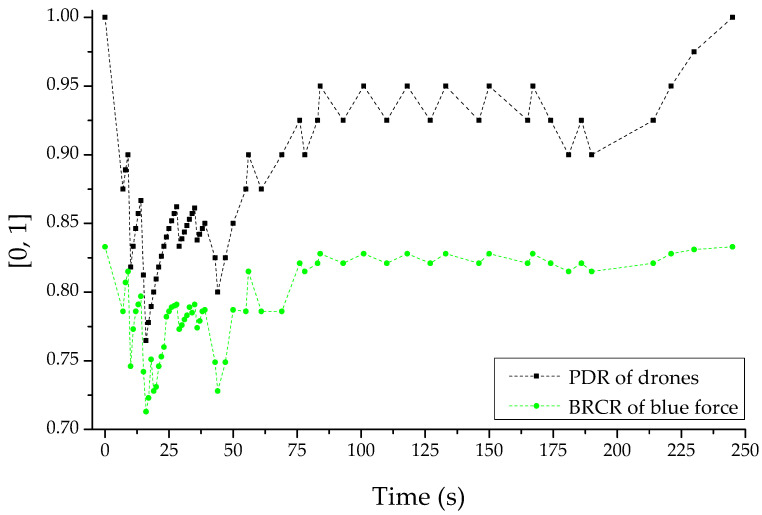
PDR (packet delivery ratio) and BRCR (blue remaining combat ratio)-based graph with channel hopping-based anti-jamming about attacker’s jammer.

**Figure 17 sensors-22-03147-f017:**
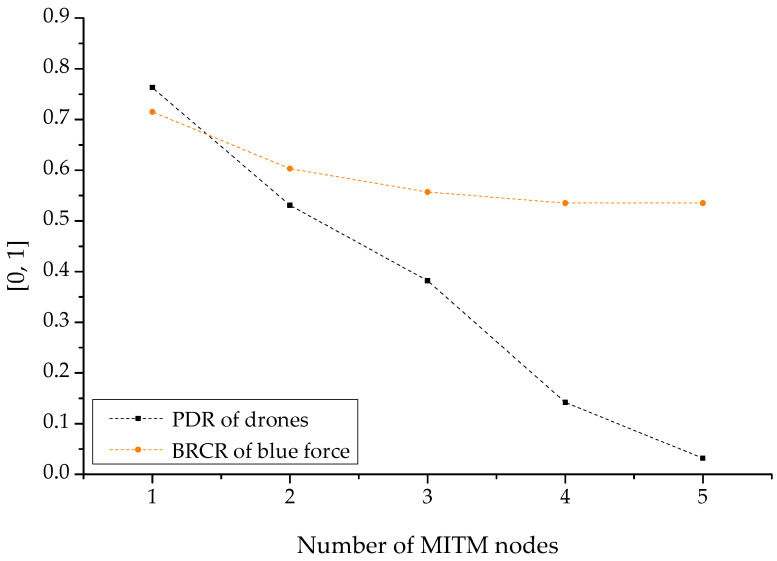
PDR (packet delivery ratio) and BRCR (blue remaining combat ratio)-based graph by MITM.

**Figure 18 sensors-22-03147-f018:**
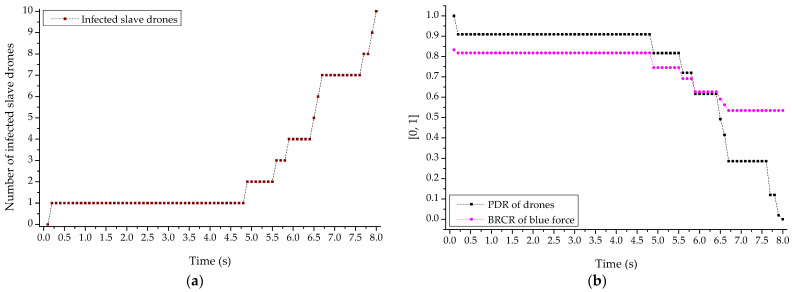
PDR (packet delivery ratio) and BRCR (blue remaining combat ratio) graph by network worm propagation and infection. (**a**) Graph of infected drones according to network worm, (**b**) PDR (packet delivery ratio) and BRCR (blue remaining combat ratio) graphs according to the increase in infected drones.

**Figure 19 sensors-22-03147-f019:**
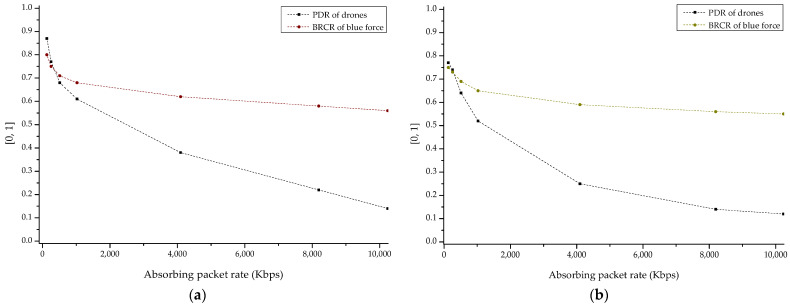
PDR (packet delivery ratio) and BRCR (blue remaining combat ratio) graph by ad hoc based routing attack with absorbing packet rate: (**a**) blackhole attack and (**b**) wormhole attack.

**Figure 20 sensors-22-03147-f020:**
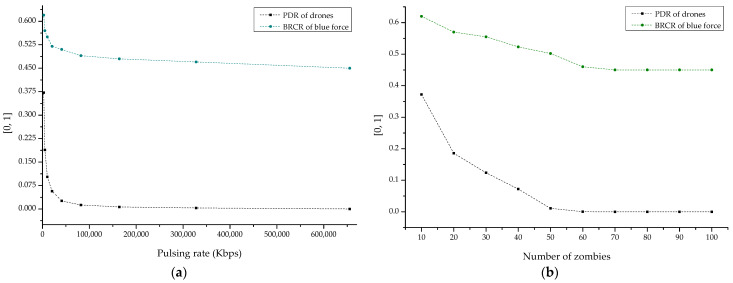
PDR (packet delivery ratio) and BRCR (blue remaining combat ratio) graph by DDoS: (**a**) pulsing rate-based and (**b**) distributed zombie-based.

**Table 2 sensors-22-03147-t002:** D-CEWS (DEVS-based cyber-electronic warfare)-based communication and functional simulation parameters for multi-layered jamming and related anti-jamming with electronic warfare threat.

Parameter (1/2)	Value	Parameter (2/2)	Value
Number of jammers	1~4	Energy supply of jammer (J)	50~100
Jammer type	Reactive	Response coefficient of jammer (%)	20~100
Initial jammer power (W)	0.1~0.5	Update period (s)	0.005~1.0
Maximum permissible jammer power (W)	0.5~1.0	Radiation angle (°)	5~60 (directivity)
Preparation time for radiation (s)	7	Response time for anti-jamming (s)	20

**Table 3 sensors-22-03147-t003:** D-CEWS (DEVS-based cyber-electronic warfare)-based communication and functional simulation parameters for MITM, spoofing, DDoS, hole-based routing attack, and network worm with cyber warfare threat.

Types	Major Parameter	Value
MITM with MAC spoofing	Number of MITM nodes	1~3
	Number of fake MAC address pair	0~3
	Probability of sniffing (%)	50~100
	Signal probability of spoofing (%)	70~100
	Probability of theft (%)	10~30
DDoS	Bulk payload (byte)	1000~500,000
	Burst pulsing rate (Kbps)	2560~655,360
	Delay time of DoS (s)	0.001~1.0
	Number of zombies	10~100
Blackhole attack	Signal probability of false routing information (%)	30~100
	Probability of masquerade (%)	20~80
	Probability of perturbation (%)	20~80
Wormhole attack	Number of collaborative attackers with tunnel	2~5
	Probability of collaborative masquerade (%)	5~50
	Probability of collaborative perturbation (%)	5~50
Worm propagation and infection	Worm model	SIR-based UDP worm
	Number of scan rate each infected node	100~500
	Payload (byte)	32~1024
	Vulnerability (%)	10~100
	Probability of infection (%)	10~90
	Number of interconnected drones	0~4
	Number of internal components in target	4~32
	Number of links between components inside the drone	4~32

**Table 4 sensors-22-03147-t004:** D-CEWS (DEVS-based cyber-electronic warfare)-based battlefield simulation parameters for combat modeling each unit in blue team force and red team force.

Battlefield		
1000 × 1000~3000 × 3000 Plane map in NS-3		
**Combat unit**		
Blue force	Tanks and infantry	10, 10
	Squad commander	1
	Drones	11
Red force	Ground vehicles and infantry	5, 5
	Cyber-electronic warfare agents	1
**PoD and PoH in combat scenario**		
Blue force	POD of blue force (%)	−0.15dis+100
	POH of blue force (%)	−0.25dis+100
Red force	POD of red force (%)	−0.1dis+100
	POH of red force (%)	−0.2dis+100

*dis* = distance between blue units and red units.

**Table 5 sensors-22-03147-t005:** Damage effect elements about targeted swarming drone system by cyber-electronic warfare threat in D-CEWS.

Threat Type	Functional Damage Effect in Battlefield							
	Message Interchange				Communication Relay	Movement	Detection	Fire
	Transmission	Receive	Eavesdropping of Sender	Eavesdropping of Receiver				
Multi-layered jamming	X	X	O (when eavesdropping jammers are used, △)	O (when eavesdropping jammers are used, △)	X	▲	▲ (partial)	O
MITM with MAC spoofing	O	X	△	△	△	▲	O	O
DDoS	X	X	O	O	X	▲	▲ (partial)	O
Blockhole attack	O	▲	△	△	▲	▲ (partial)	O	O
Wormhole attack	O	▲	△	△	▲	▲ (partial)	O	O
Worm propagation and infection	▲ (partial)	X	O	O	X	▲	O	O

O: not applicable, ▲: limit, △: eavesdropping, X: paralyzed.

## Data Availability

Not applicable.

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
