# Peer review of "D-CEWS: DEVS-Based Cyber-Electronic Warfare M&S Framework for Enhanced Communication Effectiveness Analysis in Battlefield"

_sensors, 2022, doi:10.3390/s22093147_

Round 1

Reviewer 1 Report

The following should be noted and corrected accordingly:

1. How practicable is your proposed model in real-time? 

2. Is it cost-efficient?

3. Some diagrams and terms are not properly explained.

4. Grammar is not up to standard and requires some re-editing

5. Are the formulas and numbers here generic or generated by you?

6. Paper is unnecessarily long and littered with too much irrelevant information

7. The problem it tries to solve isn't clearly explained in the Abstract

Study and consider the following related paper to embellish your paper:

• doi.org/10.3390/electronics10172110.

Revisions are required.

Author Response

Dear Reviewer

Thank you for your opinion on our paper.
We did our best to respond to your constructive recommendations.
Please refer to the attached file for more information.

Thank you.

Reviewer 2 Report

The paper proposed a framework to model and simulate the attack of defense of  the communication network used by military unmanned aerial vehicles in the battlefield.

In terms of merits, the paper provided a comprehensive study of the attacks studied and the countermeasures, together with simulation results demonstrating the effects of such attacks and countermeasures in terms of the packet delivery ratio and fraction of defender nodes remains functional.

In terms of weaknesses, a major issue is the use of language.

There are many long sentences in the paper with many words putting together, making it hard for readers to follow or understand.
Some sentences are also hard to understand, a few examples are listed below,
1. Line 158 "To alleviate and solve the above-mentioned limitations" - it is unclear to me what limitations are mentioned in Sections 2.1 and 2.2.
2. Line 165 "In addition, with regard to cyber security subjects, studies have been performed based only on general domains that are unlikely to be realized in an actual military communication environment, such as malware" - however, one of the studied attack, worm attacks, is clearly an example of malware.
3. Line 172 "It was also determined whether operational efficiency was enhanced or reduced as a result of communication issues per threat type by modelling this with the Confidentiality, Integrity, and Availability (CIA) concepts." - what was also determined?
4. Lines 176 - 180 "The adaptive countermeasures of friendly blue defenders against various wireless threats such as jamming, man-in-the-middle attack and spoofing, DDoS, black hole and wormhole-based routing attacks, and network worm propagation-infection threats were also standardized as zero-sum based anti-jamming in the case of jamming" - are all the threats being standardized as zero-sum games or only the jamming attack?
5. Lines 238 - 239 "4. Construction of Cyber-Electronic Warfare Environment in D-CEWS." - I suppose this is a typo, right?
6. Lines 278 - 281 "?_?_? defines is the defender’s detection and defense, or false-negative actions against ?_? defined as transition relations, and ?_?_? defines all actions such as reconnaissance and search for the attack point of the opposing force on ?_? Threat performance and achievement of invasion as transition relations." - I cannot quite follow here, a revision of sentence is necessary.
7. Line 317 "calculated as utility, cost, and reward" - except reward, it seems to me that utility and cost has not yet clearly defined.
8. Lines 327 - 329 "The entry into the corresponding equilibrium state was controlled according to the configuration of ?? or ?? based on whether the signaling leader was selected or not." - not sure what does it mean by "signaling leader was selected or not".
9. Lines 352 - 353 "These entities can be utilized according to the characteristics of the role and mission of the drones and can also possess structured communication processes in detail by domain" - not sure what does it mean by "in detail by domain".
10. Lines 374 - 376 "Since drones rely on wireless communication networks, threats such as hijacking, service availability and data integrity damage, privilege escalation, and remote code execution may occur if security flaws are exposed or detailed devices occupy the network" - what does it mean by "detailed devices occupy the network".
11. Lines 438 - 440 "Besides, since real-time cyber-electronic warfare threats in the wireless ad-hoc-based military unmanned aerial vehicle system were simulated." - it seems the sentence is incomplete.
12. Lines 442 - 443 "... so that they exist clearly as opposed to the existing wired legacy environment" - do you mean they are simply "complete different from the wired legacy environment"?
13. Line 697 "... they changed..." - but it seems to me that most of the parameters are not changed, right?
14. Lines 719 - 722 "When the combat power of the opposing force was decreased by more than 70% compared to the initial combat power of the opposing force and the remaining combat powers at the time of ending were formulated with experimental results. " - this sentence seems incomplete to me, or the removal of "and" in the sentence may help.
15. Lines 743 - 746 "This trend was based on all of the shutdown and availability damage of the friendly reconnaissance report communication channel due to the strong jamming power, or the loss and integrity damage of the transmitted and received reconnaissance information due to the somewhat weak jamming power." - have both situations showed in Figure 18?
16. Lines 768 - 769 "The trend was cyber agility [26]-based evasion concept called channel hopping applied within the anti-jamming response logic." - do you really mean the "trend" or the "counterattack technique"?

And the paper also has a number of "choice of word" issues, for example,
1. We usually refer to "breach" of C/I/A instead of "damage" (e.g. line 672)
2. We usually use "in the future" (e.g., lines 183, 400, and 429) to mean something not yet done, at least in the paper, instead of something to be mentioned in later parts of the paper.
3. Line 288 - as R is the reward, we will not refer it as "R's reward"

Author Response

(The authors gave the same response as above.)

Reviewer 3 Report

The idea of this paper is interesting although the following issue should be considered.
For the sake of more simplicity, the abstract should be rewritten.
the introduction consists of several subsections. I suggest summarizing the introduction and also dedicating a section of the paper to the problem statement.
I strongly recommend proofreading for whole parts. sections strat with subsections and there are no introductory for the content of each section. 
Add a paragraph to the end of the related work section and study the position of the proposed study among other studies reported in the literature.
some useless figures such as one that classifies the jammer and some that are not related to the proposed algorithms should be removed. please take care of captions and do not add redundant information to them(figure 2).
section 4 which proposes the main novelty lack an introductory and I couldn't follow the section.

Author Response

(The authors gave the same response as above.)

Round 2

Reviewer 3 Report

please consider the following issues; 
in the abstract section, writing some clues about future works is not common. abstract is too long and i think some summarizations are required. after a section, a paragraph about the content of the section is required. I mentioned this issue in my previous review but i didn't see any effect on the text. please correct it, especially for the introduction and also the proposed method.
 the references need more modification

Author Response

Thank you for your constructive and active review.

We revised the overall part of the paper to reflect your review comments as much as possible.

Please refer to the attached "rebuttal comment" document.

Thank you.
